# Amino Acids, B Vitamins, and Choline May Independently and Collaboratively Influence the Incidence and Core Symptoms of Autism Spectrum Disorder

**DOI:** 10.3390/nu14142896

**Published:** 2022-07-14

**Authors:** Laurel Jennings, Raedeh Basiri

**Affiliations:** 1Department of Nutrition and Food Studies, George Mason University, Fairfax, VA 22030, USA; ljennin6@gmu.edu; 2Institute for Biohealth Innovation, George Mason University, Fairfax, VA 22030, USA

**Keywords:** autism, autism spectrum disorder, ASD symptoms, ASD incidence, choline, vitamin B, folate, B12, B6, amino acids

## Abstract

Autism spectrum disorder (ASD) is a developmental disorder of variable severity, characterized by difficulties in social interaction, communication, and restricted or repetitive patterns of thought and behavior. In 2018, the incidence of ASD was 2.4 times higher than estimated in 2000. Behavior and brain development abnormalities are present in the complex disorder of ASD. Nutritional status plays a key role in the incidence and severity of the core symptoms of ASD. The aim of this study was to review the available peer-reviewed studies that evaluated the relationship between amino acids, choline, B vitamins, and ASD incidence and/or severity of symptoms. Through examining plasma profiles, urine samples, and dietary intake, researchers found that low choline, abnormal amino acid, and low B vitamin levels were present in children with ASD compared to those without ASD. The evidence supports the need for future research that implements simultaneous supplementation of all essential nutrients in individuals with ASD and among prenatal mothers. Future evidence could lead to scientific breakthroughs, ultimately reducing the rates of ASD incidence and severity of symptoms by applying nutritional interventions in at-risk populations.

## 1. Introduction

In the United States, 1 in 44 children were identified with autism spectrum disorder (ASD) in 2018, a major increase from the estimated 1 in 150 in 2000 [1]. Autistic disorder, Asperger syndrome, and pervasive developmental disorder can be categorized under ASD. Currently, there is no known cause for ASD; however, brain scans show that it is likely to be caused by abnormalities in brain structure and function [2]. ASD can result in restrictive, repetitive, and stereotypical behavior patterns and cause impairments in social interaction and verbal and nonverbal communication. Children with ASD experience fluctuations in aggression, hyperactivity, and attention symptoms. In addition to the symptomatic factors, ASD is costly and is a lifelong demand to both individuals with autism and their caregivers. The standard care process for those living with ASD typically involves full-time behavioral and educational therapy [3]. On average, over the individual’s lifetime, medical costs for individuals with ASD are estimated to be about USD 293,545 more expensive when compared with an individual who developed typically. Thirteen years of special education for those with ASD may cost on average USD 37,872 more than education for a typically developing student. Moreover, lifetime care costs (medical and at-home care) are estimated at around USD 967,493 for an individual living with ASD. In addition, over the course of the individual with ASD’s lifetime, the family may experience a loss of productivity, costing around USD 2,109,358 [3]. Evidence shows that individuals with ASD frequently suffer from disrupted nutrient levels [4]. It has been shown that supplementation with specific nutrients can decrease the incidence of ASD and alleviate the severity of symptoms. Nutritional interventions before and during pregnancy as well as during early ages are a feasible and cost-effective way of preventing and controlling symptoms of ASD. The aim of this study was to examine the effects of nutritional status and supplementation on the incidence and or severity of ASD symptoms using currently available resources.

## 2. Materials and Methods

A comprehensive literature review was conducted to analyze currently available evidence on the relationship between AAs, choline, B vitamins, and ASD incidence and/or severity of symptoms. Inclusion criteria included English peer-reviewed articles that were published between 2010 and 2022 examining the effects of choline, B vitamins, and amino acids (AA) levels and supplementation on the incidence and/or symptoms of ASD. Searches used databases, Primo through the George Mason University Library, PubMed, and Google Scholar. The following search terms and sequences were used within the databases to improve search effectiveness: “ASD” or “autism” or “choline” or “vitamin B” or “B12” or “B6” or “folate” or “amino acid” or “symptom” or “incidence” or “occurrence”. A final webwide search was used to identify any peer-reviewed studies existing outside of the identified databases. Moreover, the reference sections for the studies found were sorted through to find additional research studies. A total of 132 articles were identified. Searches were filtered to “reviews” to locate the current state of science and prevent duplication. We identified sparsity in review papers that evaluate the effects of choline, B vitamins, and AAs on the incidence and symptoms of ASD. Therefore, these nutrients were chosen to take a deeper look into their impact individually and collectively on the incidence and symptoms of ASD. Searches were then filtered to “articles” to find the available peer-reviewed articles on choline, B vitamins, and amino acids. After multiple rounds of screening, 15 articles were eligible to be included in this study.

## 3. Results

Adequate nutrition is necessary for optimizing brain function and preventing cognitive disorders. B vitamins, choline, certain amino acids, vitamin D, and omega 3 show neuroprotective effects and play a key role in improving intellectual performance [5]. Evidence showed that the study of choline, amino acids, vitamin B6, vitamin B12, and folate is newer in ASD than other nutrients such as vitamin D and omega 3. Therefore, this review will focus on the effects of choline, amino acids, and B vitamins on the incidence and symptoms of ASD.

### 3.1. Choline

Choline is typically taken into the body through diet. It is an essential nutrient and plays an important role in neurotransmitter synthesis (acetylcholine), methyl-group metabolism (homocysteine reduction), cell-membrane signaling (phospholipids), and lipid transport (lipoproteins) [6,7]. Choline is also known to affect sensory processing, cognitive functioning, memory, and learning which are often atypical in individuals with ASD [8]. It is known that choline contributes to brain development [9] and assists in the production of methionine, an essential amino acid [10]. Hamlin et al. evaluated choline and betaine effects in children with and without ASD by assessing their dietary intake and blood levels [11]. They showed that the children with ASD had a dietary intake of choline that was below the dietary reference’s intake level appropriate for their age. Moreover, plasma levels of choline were significantly lower in a subgroup of children with ASD when compared to the age-matched control children without ASD [11].

Individuals with ASD often experience impaired functioning of the central nervous system (CNS) as well as metabolic disorders [12]. Choline’s role in the synthesis of acetylcholine and reduction of homocysteine can help with improving core symptoms of ASD [13]. To evaluate the effects of choline in improving language and core ASD symptoms in children, Gabis et al. conducted a nine-month randomized, double-blind, placebo-controlled trial following 60 children with ASD [6]. They examined choline supplementation (350 mg) alongside donepezil (5 mg), a prescription drug that inhibits the breakdown of acetylcholine (ACh) [4]. This study aimed to increase ACh activity in the synaptic cleft by increasing ACh in the brain via supplementing with choline and preventing its breakdown by giving patients donepezil. Researchers found the combined treatment to improve receptive language skills after 12 weeks of treatment, primarily in young children (10 years and younger). Improvement stayed consistent even six months after treatment. This study reported no side effects for the use of the regimen in children aged under 10 years; however, those above the age of 10 showed some worsening in behavior, specifically with irritability [6].

These effects have been shown in animal studies as well. Agam et al. studied methylenetetrahydrofolate reductase (MTHFR)-deficient mice to resemble the common gene abnormalities associated with an increased risk for ASD [10]. The MTHFR enzyme assists in processing amino acids; particularly, it is important in the conversion of homocysteine to methionine due to its role in the metabolism of folic acid [14]. This gene abnormality has been shown to be present in both mothers and their offspring with ASD. The offspring of MTHFR-deficient mice which received choline supplemented drinking water (0.003%) for two weeks showed a reduction in characteristics related to repetitive behavior and anxiety. Additionally, in male mice, social behavior and abnormal cortical protein levels of autophagy markers (*p62* and *Beclin-1*) were improved [10]. When compared to controls, both up- and down-regulation of autophagy have been associated with autism [15,16,17,18]. One study finds autophagy marker, beclin-1, to be decreased for both males and females, and LC3 to be increased for females and decreased for males [15]. Another study found the autophagy regulator, mTOR, to be overactive in those with ASD-like behaviors [16]. This association can be understood because autophagy plays a role in the brain development of humans, and normal autophagy is associated with the prevention of neurodevelopmental disorders, such as ASD [19].

Adequate intake of choline is also essential during gestation, as it contributes to brain development. An inadequate amount of choline could also influence the brain development of those with ASD, resulting in symptoms being more severe. A study evaluated the impact of choline consumption during pregnancy and nursing to evaluate social interaction and anxious behaviors [9]. Social behavior, anxiety, and repetitive behaviors pre- and post-choline supplementation was examined in a particular mouse strain that displays autism-like phenotype behavior. Through analysis, researchers found choline supplementation to reduce deficits in social interaction, lower anxiety levels, and reduce marble-burying behavior in mice [9]. Marble-burying is an animal model used in scientific research to depict anxiety or obsessive-compulsive disorder (OCD) behavior [20].

The reported benefits of choline supplementation may be partially due to its role in the improvement of potassium, calcium, and sodium chloride ions transportation. Olson et al. studied the potential positive effects of dietary choline intake on improving sensory processing function in ASD [8]. The established idea that acetylcholine supports ion transport in the body was used to suggest that proper intake of choline through the diet would increase acetylcholine and sequentially increase ion transport in the body, which would improve sensory processing in ASD.

### 3.2. B Vitamins

B vitamins are taken into the body through diet and supplementation, including fortification and enrichment. B vitamins are essential to many of the body’s processes involving the CNS, oxygen transportation, blood cell production, and amino acid production and conversion [21,22,23]. Folate is known to assist in converting homocysteine to methionine [10]. Vitamin B6 contributes to the conversion and degradation of amino acids via the transfer of nitrogen. It also contributes to the production of neurotransmitters (serotonin and dopamine), glutathione, and hemoglobin [23]. Earlier research suggested that vitamin B6 and magnesium supplementation may result in improved ASD symptoms [24,25,26]. Each of these vitamins plays a distinctive role in CNS function. Deficiency of folate may cause behavior changes and cognitive impairment, while deficiency of vitamin B6 can result in irritability [27]. The deficiency of vitamin B12 has features of neurological impairments such as motor disturbances, cognitive impairment, irritability, and brain cell loss, all commonly known as symptoms of ASD [21,27]. 

Evidence has shown that adequate intake of B vitamins is important in preventing behavioral and cognitive disorders, significant concerns in those with ASD. Schmidt et al. examined the associations between autism and maternal vitamin supplement intake during the periods of preconception and prenatal development. They recruited 545 children between 24 and 60 months of age from the large, population-based, case-control study, Childhood Autism Risks from Genetics and Environment (CHARGE) [28]. Children were grouped based on diagnoses and their cognitive function and assessed using validated questionnaires, behavioral scales, and learning scales. Mothers answered questions specific to vitamin supplementation and fortification intake at any time during three months before conception, through pregnancy, and during the period of breastfeeding. The prenatal vitamins used by mothers typically contained iron, vitamin B6, vitamin B12, and folic acid (800 μg). Findings of this study showed that prenatal vitamin intake during the three months before conception and the first month of pregnancy was associated with a reduced risk for autism. Additionally, researchers studied vitamin intake in normal participants and those with abnormalities in folate, methionine, and transmethylation pathways. Genotyping was determined through blood collection from all family members. Through genetic testing, folate-related pathways were found to be more abnormal in those who were genetically susceptible to developing autism [28]. 

Another study by Raghavan et al. aimed to determine if multivitamin supplementation during pregnancy and maternal levels of plasma folate and B12 had an association with the incidence of ASD [29]. Through the Boston Medical Center, researchers recruited and reported on 1257 mother–child pairs that were followed from birth throughout childhood. Mothers who reported multivitamin supplementation from three to five times per week were found to have a lower chance of birthing a child with ASD. Those supplementing less than three or more than five times per week were found to have a higher chance of birthing a child with ASD. Very high blood levels of folate (>2.2 micrograms per deciliter) and B12 (>19.5 micrograms per deciliter) were associated with having a two and a half times higher risk of birthing a child with ASD [29]. Similarly, Steenweg-de Graaff et al. examined the association between human folate concentrations during pregnancy and the severity and presence of autistic traits in their offspring during six years after birth in a population-based birth cohort in the Netherlands [30]. Maternal weight, age, and previous pregnancies were taken into consideration and excluded where appropriate to reduce exposure to other risks known to increase the chances of birthing a child with ASD. Plasma folate levels were taken amongst 5591 mothers in early pregnancy between 10.5- and 17.2-week gestation. At the children’s age of six, researchers were able to obtain information on the autistic traits of 70% of the cohort. After analysis and adjustment for confounders, researchers did not find folic acid supplementation to be protective at a significant level in lowering the chances of birthing a child with autistic traits [30]. In contrast, another study with a larger population (*n* = 85,176) found folic acid to decrease the incidence of ASD. Surén et al. studied the association between folic acid supplementations prior to pregnancy and the risk of children developing ASD [31]. Children aged 3–10 years were studied in a population-based, prospective cohort study in Norway. This study attained information on folic acid intake four to eight weeks after the start of pregnancy in the mothers of the studied children. Of the 85,176 children studied, 114 were diagnosed with ASD after following up between the ages of 3–10 years. Of those diagnosed with ASD, 64 mothers were supplementing with folic acid, while 50 were not. Interestingly, after adjustments for demographics, those with Asperger syndrome and pervasive developmental disorder did not show an association between mothers’ folic acid intake and ASD incidence [31].

In a clinical trial of 57 children aged 3–7 years, Hendren et al. directly studied the impact that 75 micrograms (about 63% more than the recommended dietary allowance for this group) of vitamin B12, in the form of methylcobalamin, would have on improving symptoms in children with autism [32]. The study aimed to improve the methylation of methionine and metabolism of the antioxidant glutathione, which were measured at baseline and after eight weeks of treatment. Results showed that methylcobalamin treatment improved ASD symptoms. These findings were correlated with improvements in transmethylation metabolism, which is known to be abnormal in individuals with ASD [32]. 

B vitamins also influence oxidative stress and inflammation in the body by assisting in antioxidant reactions. Wang et al. studied the relationship between B vitamins supplementation and autism-like behavior and neurodevelopmental impairment in an animal study [33]. They showed that supplementation with folate, B6, and B12 significantly reduced neurobehavioral impairment in autistic mice, including reduced social communication disorder, reduced stereotyped repetitive behavior, and reduced learning and spatial memory impairment. Moreover, reductions in mitochondrial damage, pro-inflammatory cytokines, and increases in gene activities that assist in the synthesis of superoxide dismutase, glutathione, and glutathione peroxidase were observed. In addition to the B vitamin’s ability to reduce oxidative stress, they can also lower the plasma concentration of homocysteine, which adds to the antioxidant capabilities of these vitamins by further reducing oxidative stress and inflammation [34].

B vitamins, particularly B6, B12, and folate, influence both the central and peripheral nervous system’s function by contributing to maintaining a healthy nervous system and improving neurological conditions, even when a deficiency is not determined [35,36]. The effects of B vitamins on CNS function can be partly due to their roles in metabolizing amino acids. Kałużna-Czaplińska et al. showed that supplementation with vitamins B12, B6, and magnesium would better stabilize urinary tryptophan concentration in children with ASD [12]. Urinary tryptophan levels in the supplementation group were from 0.07 to 19.67 μmol/mmol, while in the control group, they were from 0.01 to 348.94 μmol/mmol [12]. Along with other AAs, tryptophan serves as a precursor for many major neurotransmitters, especially serotonin [12]. Therefore, controlled urinary tryptophan levels might show that tryptophan was used for the synthesis of neurotransmitters in this population. 

### 3.3. Amino Acids

Essential amino acids must be consumed in the diet, while non-essential amino acids can be created by conversion processes in the body. Amino acids are known to have a significant impact on the CNS, controlling the body and mind [37,38]. Certain amino acids play an important role in regulating CNS neurotransmitters, including serotonin and dopamine. Tryptophan is known to increase serotonin levels, while tyrosine increases dopamine levels [39,40]. Serotonin is involved in brain development, as it influences cell division, cell proliferation, migration, differentiation, cortical plasticity, and synaptogenesis [41,42]. Moreover, serotonin influences memory, learning ability, and mood [43,44]. Dopamine regulates motor activity, motivation, attention, and reward processing [45,46,47]. Individuals with autism have impaired dopamine and serotonin levels [48,49]; therefore, adequate levels of tryptophan and tyrosine might support normal concentrations of dopamine and serotonin. Inadequate levels of tyrosine have been shown to decrease focus and increase hyperactivity in children without ASD [50,51]. 

Differences in the plasma levels of amino acids amongst individuals with ASD and individuals without ASD have been documented. Table 1 reports the differences found in the reported amino acid plasma levels between individuals with ASD and individuals without ASD. Individuals without ASD are shown to have normal levels across the plasma levels of the listed amino acids, while individuals with ASD exhibit a range of high and low levels. 

High levels of amino acid lysine have been supported via examining plasma levels [52,55]. Although infrequent, high levels of essential amino acid lysine have been associated with an intellectual disability or behavioral issues [59]. Deficiencies in amino acid lysine among children with ASD have also been supported by earlier research [54]. Inadequate levels of essential amino acid lysine can cause agitation [38,60]. This might be due to a lack of synthetization of glutamate, which is responsible for producing the neurotransmitter GABA [61]. GABA is the main inhibitory neurotransmitter, and a low level of GABA is associated with mood disorders [62]. 

Arum et al. evaluated phenylalanine and tryptophan intake amongst hyperactive children with ASD to evaluate the association between amino acid levels and the presence and severity of hyperactivity in individuals with ASD [38]. Researchers found elevated tryptophan and phenylalanine levels to be associated with increased hyperactivity. The elevated levels of phenylalanine could be a result of an inadequate conversion of this amino acid to tyrosine, which raises concerns that there may have also been inadequate levels of tyrosine in these children [38]. As multiple research studies suggest, tyrosine was lower in the plasma levels of those with ASD [52,53,54]. Low levels of tyrosine can contribute to ASD symptoms, such as the inability to concentrate, focus, and increased levels of hyperactivity [38,63]. Low levels of tryptophan have also been found [52,53,54]. As tyrosine increases dopamine levels in the brain and tryptophan increases serotonin levels in the brain, this lack of tyrosine and tryptophan in ASD can cause an imbalance between dopamine and serotonin levels leading to more severe ASD symptoms. Moreover, a late diagnosis of high phenylalanine levels is likely to increase the chances of autism and related symptoms. Implementation of a phenylalanine-free diet may improve ASD symptoms in those with elevated levels. It is important to note that the population of this study did not evenly account for the various levels of hyperactivity. Additionally, they were not compared to individuals without hyperactivity [39]. A study by Saad et al. [37] examined the relationship between a late diagnosis of PKU and the development of ASD symptoms. They measured the presence of autistic symptoms among 32 children who were diagnosed with phenylketonuria (PKU) after one year of age. Among the 32 children diagnosed with late-stage PKU, eight (25%) were diagnosed with autism (females *n* = 3 and males *n* = 5). Two of them had severe autism, and six had mild/moderate autism. The two children diagnosed with severe autism were both male [37]. Demirci et al. showed that when PKU and ASD co-present in a child, following a phenylalanine-free diet helps with improving ASD scoring and symptoms such as eye contact, awareness, and word formulation [64]. 

These findings show that although amino acids are crucial for the proper function of the CNS, overconsumption of these nutrients might disrupt the normal function of the nervous system. For instance, toxic levels of essential amino acid phenylalanine have a neurotoxic effect, damaging the CNS [65]. It has been shown that toxic levels of phenylalanine significantly increase the incidence of ASD [64] and could lead to behavioral problems and intellectual disabilities [37]. Therefore, it is important to consume optimum levels of amino acids to maximize their positive effects while preventing their negative effects on CNS function. 

Not only do amino acids and the CNS interact, but B vitamin levels may also interact with amino acids. High concentrations of non-essential amino acid homocysteine may indicate low levels of vitamin B12, B6, and folate [66]. Inadequate amounts of B vitamins, such as B12 and folate, decrease the conversion of homocysteine to the essential amino acid, methionine. This buildup of homocysteine has been associated with cognitive impairment and various psychiatric disorders [67]. Al-Farsi et al. found that folate and B12 levels were lower in children who were newly diagnosed with autism [68]. This study followed 80 children with and without ASD and showed that children with ASD had a lower dietary intake of folate and vitamin B12 compared to those without ASD. Homocysteine levels were 68% higher, and methionine levels were 15% lower in the ASD group, which could support the conversion error in those with B12 and folate deficiency. In another study, Liu et al. examined the urinary amino acids present in 57 children younger than 14 years old with ASD and 82 healthy children with typical development (TD) [63]. Proper measures were taken to exclude children that had other mental illnesses, severe neurological diseases, and those taking supplements during specimen collection. The analysis found 44 urinary AAs, 10 urinary AAs indices, and 9 urinary AAs-related metabolites. Among the 63 urinary AAs indicators present, 16 AAs indicators were significantly different in children with ASD when compared with TD children, with 9 AAs indicators being significantly higher: methionine sulfoxide, homoarginine, 3-methyl-histidine, creatine, arginine, arginine/ornithine ratio, ornithine/citrulline ratio, 5-hydroxytyrptamine, and 4-hydroxyproline. Compared with TD children, researchers found children with ASD to poorly excrete amino acid homocysteine from the body.

In ASD, B vitamins and amino acids have a significant relationship since B vitamins influence the metabolism of amino acids that regulate CNS functions [66]. B vitamin deficiencies decrease the re-methylation of AA homocysteine [69]. Yektaş et al. investigated the serum concentrations of amino acid, homocysteine, vitamin B12, and folate in children who were either diagnosed with ASD or attention deficit hyperactivity disorder (ADHD) and compared them to healthy controls [66]. Moreover, the severity of symptoms was assessed among individuals with ASD to assess the relationship between serum concentrations of the aforementioned nutrients and ASD symptom severity. Both male and female individuals with ASD had the lowest vitamin B12 and the highest homocysteine levels compared to the ADHD group and the healthy control group. The low vitamin B12 levels were correlated with increased hyperactivity and/or impulsivity and oppositionality symptoms among children with ASD. Folate levels did not show significant variations. 

### 3.4. Nutrient Roles

Table 2 summarizes current evidence regarding the examined nutrients’ confirmed roles in cognitive and brain function as well as their roles in ASD incidence and severity of symptoms. 

### 3.5. Interrelationship between B Vitamins, Choline, Amino Acids, and Neurotransmitters

Figure 1 shows how folate, vitamin B12, vitamin B6, and choline aid in amino acid conversion processes, neurotransmitter synthesis, and cell-membrane signaling, which are all important in normal cognitive and brain function. Adequate consumption of folate, B12, and choline helps to convert homocysteine into methionine, while vitamin B6 is necessary for converting homocysteine into cysteine. The conversion of homocysteine to methionine then encourages neurotransmitter synthesis (acetylcholine) and cell-membrane signaling (phospholipids). Vitamin B6 aids in the conversion of phenylalanine to tyrosine, which then facilitates the synthesis of the neurotransmitter dopamine. Tryptophan can be converted to serotonin with the aid of folate, B12, and B6. Disrupted levels of dopamine and serotonin have been reported in children with ASD [48,49]. 

## 4. Discussion

Evaluating current evidence showed that children with ASD have higher rates of abnormal amino acids and lower blood levels of choline, B6, B12, and folate when compared to those without ASD. Significantly lower dietary intake of the aforementioned nutrients has also been reported in children with ASD, which could be a reason for the observed lower plasma levels of these nutrients in children with ASD. The reviewed research suggests that homeostatic levels of amino acids, B vitamins, and choline can reduce the incidence of ASD and improve related symptoms. Findings showed that increasing dietary intake of choline could improve anxious behaviors, receptive language skills, social behavior, sensory processing, and other symptoms which rely on ion transport in individuals with ASD. In combination and at appropriate levels, consumption of listed B vitamins could also reduce the risk of developing ASD and improve neurobehavioral impairment. On the other hand, it is important to note that while these vitamins have been reported to reduce the risk of ASD when levels are met appropriately, intake of these vitamins at toxically elevated levels could increase the risk of developing ASD.

In children with ASD, abnormalities in amino acid levels could be due to inadequate intake or poor metabolism. Choline and B vitamins play important roles in improving metabolism and maintaining homeostatic plasma and urinary levels of amino acids; thus, supplementation with these nutrients leads to improvements seen in ASD symptoms. More specifically, vitamin B6, -B12, and choline may work collaboratively to improve the homeostasis of essential amino acids, tryptophan, tyrosine, and phenylalanine. Findings suggest elevated levels of tryptophan and phenylalanine may negatively influence symptoms of ASD [38,64]. Elevated levels of phenylalanine might occur as a result of the lack of conversion of phenylalanine to tyrosine. This could lead to increased hyperactivity and decreased focus due to inadequate available tyrosine, which plays an important role in the production of dopamine. This evidence supports the need for future research that implements interventions with dietary amino acids, choline, and B vitamins in individuals with ASD and in pregnant women at high risk of birthing a child with ASD to confirm the suggested positive effects. 

Despite the potential relationship between choline, B vitamins, and amino acids and the incidence of ASD as well as the severity of symptoms related to ASD, the synergistic effects of supplementation with all of these nutrients in individuals with ASD has not been studied yet. To our knowledge, this is the first review evaluating existing evidence on the potential relationship between amino acids, B vitamins, choline, and ASD incidence and severity of symptoms. More clinical trials with stronger designs and larger populations could confirm the effects of these nutrients on the incidence and core symptoms of ASD. Future studies should focus on discovering optimum levels of these nutrients for the prevention of ASD and alleviation of symptoms in children with ASD in different age groups. Altering nutritional status can be an affordable and effective way to prevent ASD and improve the quality of life for families and individuals impacted by ASD.

## Figures and Tables

**Figure 1 nutrients-14-02896-f001:**
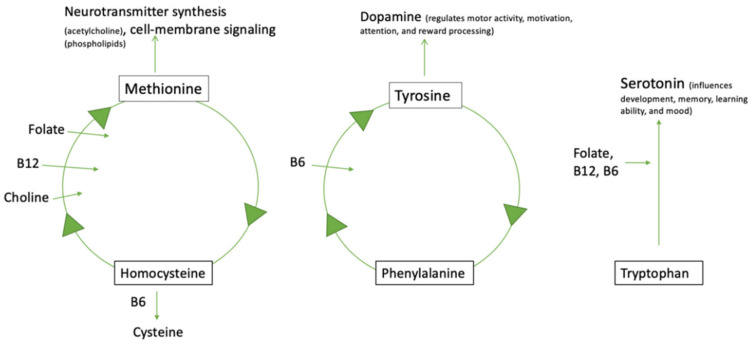
The interrelationship between choline, B vitamins, amino acids, and important neurotransmitters in ASD.

**Table 1 nutrients-14-02896-t001:** Plasma levels of amino acids in individuals with ASD compared to individuals without ASD.

Amino Acid	ASD
Tryptophan	High [38] Low [52,53,54]
Tyrosine	Low [52,53,54]
Phenylalanine	High [38,55,56]
Homocysteine	High [52,57,58]
Lysine	High [52,55] Low [54]

**Table 2 nutrients-14-02896-t002:** Nutrient Roles.

Nutrient	Confirmed Roles	Evidence-Based Findings Relevant to ASD
Choline	Converts into acetylcholine in the body [6]	Lower dietary intake and plasma levels of choline in children with ASD [11] Supplementation (36 mmol/kg) improved anxiety levels, deficits in social interaction, and reduced marble-burying behavior in mice with ASD-like behavior [9]
	Aids in the production of methionine [10]	Supplementation (350mg) resulted in improved language skills in children ≤10 years old [6]
	Contributes to brain development [9]	Supplementation (350mg) increased acetylcholine which improved ion transport resulting in ameliorated ASD symptoms [6] Supplementation decreased repetitive behavior and anxiety among ASD-induced mice [10] Supplementation improved social behavior and cortical protein levels of autophagy markers (*p62* and *beclin-1*) among ASD-induced mice [10] Improves sensory processing, cognitive functioning, memory, and learning [32]
Tryptophan	Involves in the synthesis of serotonin [40]	Elevated dietary intake increased hyperactivity among individuals with ASD [38]
Tyrosine	Involves in the synthesis of dopamine [39]	Low blood levels of tyrosine can decrease dopamine, leading to a decrease in focus and an increase in hyperactivity in ASD [38]
Phenylalanine	Converts into tyrosine [38] Toxic levels of phenylalanine have a neurotoxic effect [65] Elevated levels of phenylalanine could lead to behavioral problems and intellectual disabilities [70]	High dietary intake of phenylalanine was reported in children with ASD [38] 25% of late-diagnosed PKU children had autism [37] Well-established relationship between toxic levels and ASD [64] Phenylalanine free diet showed improvements in symptoms for children diagnosed with PKU/ASD [64]
Lysine	Deficiency leads to disruption of glutamate synthesis, which interferes with gamma-aminobutyric acid (GABA), an important neurotransmitter in CNS synthesis that reduces neuronal excitability by inhibiting nerve transmission [61,71]	Lower urinary levels among individuals with ASD [63] A lack of lysine causes agitation in children with ASD/ADHD [38,60]
Homocysteine	Elevated levels may indicate low vitamin B12, B6, and folate [66] Increased homocysteine levels are associated with decreasing cognitive function and dementia [72]	Lower urinary levels among children > 14 years with ASD and no coexisting illnesses [63] Higher serum concentrations among 81% male population of children with a median age of eight years with ASD [66]
Vitamin B6	Contributes to the production of neurotransmitters (serotonin and dopamine), glutathione, and hemoglobin [23] Aids in the synthesis and conversion of amino acids and neurotransmitters [22] Deficiency can cause irritability [27]	Reduced risk for birthing a child with autism when supplementation is paired with iron, B12, and folic acid [28] Supplementation contributed to more stable tryptophan levels [12] When supplemented with folate and B12, reduced neurobehavioral impairment (social communication disorder, stereotyped repetitive behavior, learning and spatial memory impairment) in mice [33]
Vitamin B12	Deficiency has features of neurological impairments such as motor disturbances, cognitive impairments, irritability, and brain cell loss [21,27] Blood cell production [21]	Lower dietary intake and serum concentrations were observed among children with ASD [66,68] Reduced risk for birthing a child with autism when supplementation is paired with iron, B6, and folic acid [28] High plasma levels (>19.5 μg per deciliter) during pregnancy were associated with increased risk for ASD [29] Supplementation (75 μg) correlated with improvements in transmethylation metabolism of AA methionine and improvements in ASD symptoms [32] Supplementation contributed to more stable tryptophan levels, which aids in normal production of neurotransmitters [12] Combined supplementation with folate and B6 reduced neurobehavioral impairment (social communication disorder, stereotyped repetitive behavior, learning, and spatial memory impairment) in mice [33]
Folate	Aids in converting AA homocysteine to methionine [10]	Low dietary intake was observed among children with ASD [68] No significant variations between ASD and non-ASD were measured by serum and plasma [30,66]
	Deficiency can cause behavior changes and cognitive impairment [27]	High plasma levels (>2.2 μg) during pregnancy were associated with an increased risk of birthing a child with ASD [29] A higher risk for autism was found when mothers did not supplement with folic acid before pregnancy [31] Reduced risk for birthing a child with autism when supplemented along with iron, B12, and B6 [28] Supplementation contributed to more stable tryptophan levels, which aids in normal production of neurotransmitters [12] Combined supplementation with folate and B12 reduced neurobehavioral impairment (social communication disorder, stereotyped repetitive behavior, learning, and spatial memory impairment) in mice [33]

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
