# Peer review of "Amino Acids, B Vitamins, and Choline May Independently and Collaboratively Influence the Incidence and Core Symptoms of Autism Spectrum Disorder"

_nutrients, 2022, doi:10.3390/nu14142896_

Round 1
Reviewer 1 Report
This is a well-researched review article on the nutritional roles of amino acids, vitamin B and choline in autism. The authors conducted an in-depth literature searches on relevant studies published between 2010 and 2022. Overall, the manuscript was easy to follow and understand. Each section is well-organized and without redundant information. I have a few minor comments:
1. In lines 115-116, please further describe the abnormal autophagy observed in autism. How does abnormal autophagy contribute to autism?
2. Have there been any studies reporting the plasma levels of amino acids in autistic and non-autistic patients? It would be useful to the readers to include a table outlining the plasma levels of amino acids comparing autistic and non-autistic patients.
Author Response
We would like to thank the reviewer for taking the time and effort to review our manuscript. Our responses to the Reviewer’s comments are described below point by point. Changes, suggested by the reviewer, have been incorporated into the manuscript using track changes within the document. The responses are also provided below.
- In lines 115-116, please further describe the abnormal autophagy observed in autism. How does abnormal autophagy contribute to autism? We thank the reviewer for this comment. We provided more specific information about the abnormal autophagy observed in autism in lines 116 to 123 and cited more references. Please see the paragraph below.
“ When compared to controls, both up- and down-regulation of autophagy have been associated with autism[15–18]. One study finds autophagy marker, beclin-1 to be decreased for both males and females, and LC3 to be increased for females and decreased for males[15]. Another study found the autophagy regulator, mTOR, to be overactive in those with ASD-like behaviors.[16] This association can be understood because autophagy plays a role in the brain development of humans and normal autophagy is associated with the prevention of neurodevelopmental disorders, such as ASD[19].”
- Have there been any studies reporting the plasma levels of amino acids in autistic and non-autistic patients? It would be useful to the readers to include a table outlining the plasma levels of amino acids comparing autistic and non-autistic patients. We agree with the reviewer that including a table comparing plasma levels of amino acids in autistic and non-autistic individuals would be useful to the readers. We have added this table and provided more information about the levels of amino acids in individuals with and without ASD in lines 256 to 269 as well as lines 281 to 286. Please see below.
“Differences in the plasma levels of amino acids amongst individuals with ASD and individuals without ASD have been documented. Table 1 reports the differences found in the reported amino acid plasma levels between individuals with ASD and individuals without ASD. Individuals without ASD show to have normal levels across the plasma levels of the listed amino acids, while individuals with ASD exhibit a range of high and low levels.
Table 1. Plasma Levels of Amino Acids in individuals with ASD compared to individuals without ASD
Amino Acid |
ASD |
Tryptophan |
High [38] Low [52–54] |
Tyrosine |
Low [52–54] |
Phenylalanine |
High [38,56,57] |
Homocysteine |
High [52,58,59] |
Lysine |
High [52,56] Low [54]
|
High levels of amino acid lysine have been supported via examining plasma levels[52,56]. Although infrequent, high levels of essential amino acid lysine have been associated with intellectual disability or behavioral issues[60]. Deficiencies in amino acid lysine among children with ASD have also been supported by earlier research [54]. Inadequate levels of essential amino acid lysine can cause agitation[38,61]. This might be due to a lack of synthetization of glutamate, which is responsible for producing the neurotransmitter GABA [62]. GABA is the main inhibitory neurotransmitter and a low level of GABA is associated with mood disorders [63].”
“As multiple research studies suggest, tyrosine was lower in the plasma levels of those with ASD[52–54]. Low levels of tyrosine can contribute to ASD symptoms, such as the inability to concentrate, focus, and increased levels of hyperactivity[38,64]. Low levels of tryptophan have also been found[52–54]. As tyrosine increases dopamine levels in the brain, and tryptophan increases serotonin levels in the brain, this lack of tyrosine and tryptophan in ASD can cause an imbalance between dopamine and serotonin levels leading to more severe ASD symptoms.”

Reviewer 2 ReportThe review of nutrients such as choline, amino acids and B vitamins in relation to the severity of autism as well as in the possible prevention has been adequately carried out including the necessary studies to be able to assess this relationship. It is interesting to note that excessive doses of vitamins or amino acids such as phenylalanine or tyrosine could have harmful effects aggravating autism itself.
Author Response
The review of nutrients such as choline, amino acids, and B vitamins in relation to the severity of autism as well as in the possible prevention has been adequately carried out including the necessary studies to be able to assess this relationship. It is interesting to note that excessive doses of vitamins or amino acids such as phenylalanine or tyrosine could have harmful effects aggravating autism itself.
Thank you for your feedback and positive evaluation of our study!